# Modern Methods of Diagnostics and Treatment of Neurodegenerative Diseases and Depression

**DOI:** 10.3390/diagnostics13030573

**Published:** 2023-02-03

**Authors:** Natalia Shusharina, Denis Yukhnenko, Stepan Botman, Viktor Sapunov, Vladimir Savinov, Gleb Kamyshov, Dmitry Sayapin, Igor Voznyuk

**Affiliations:** 1Baltic Center for Neurotechnologies and Artificial Intelligence, Immanuel Kant Baltic Federal University, 236041 Kaliningrad, Russia; 2Department of Social Security and Humanitarian Technologies, N. I. Lobachevsky State University of Nizhny Novgorod, 603022 Nizhny Novgorod, Russia; 3Department of Neurology, Pavlov First Saint Petersburg State Medical University, 197022 Saint Petersburg, Russia

**Keywords:** neurodegenerative diseases, depression, electroencephalography, machine learning, neural network

## Abstract

This paper discusses the promising areas of research into machine learning applications for the prevention and correction of neurodegenerative and depressive disorders. These two groups of disorders are among the leading causes of decline in the quality of life in the world when estimated using disability-adjusted years. Despite decades of research, the development of new approaches for the assessment (especially pre-clinical) and correction of neurodegenerative diseases and depressive disorders remains among the priority areas of research in neurophysiology, psychology, genetics, and interdisciplinary medicine. Contemporary machine learning technologies and medical data infrastructure create new research opportunities. However, reaching a consensus on the application of new machine learning methods and their integration with the existing standards of care and assessment is still a challenge to overcome before the innovations could be widely introduced to clinics. The research on the development of clinical predictions and classification algorithms contributes towards creating a unified approach to the use of growing clinical data. This unified approach should integrate the requirements of medical professionals, researchers, and governmental regulators. In the current paper, the current state of research into neurodegenerative and depressive disorders is presented.

## 1. Neurodegenerative Disorders

Neurodegenerative disorders are a group of slowly progressing irreversible conditions characterized by neuronal death and subsequent atrophy of certain areas of the brain. These disorders predominantly manifest in old age and include Alzheimer’s disease, Parkinson’s disease, Huntington’s disease, Pick’s disease, amyotrophic lateral sclerosis, dementia with Lewy bodies, and other similar other associated conditions [1]. All of them are characterized by a decrease in cognitive abilities, severe motor impairment, impaired social functioning, and significant difficulties in one’s day-to-day tasks. Currently, neurodegenerative disorders are not curable, and their treatments are aimed at slowing down the disease’s progression, improving the quality of life of patients, and addressing comorbid disorders. The number of people with known neurodegenerative disorders has increased over the past decades. This increase has been predominantly associated with an increase in life expectancy (so, individuals are more likely to develop a disorder during their lifetime) and an increase in the effectiveness of early diagnostics (more cases became known to healthcare systems) [2]. According to recent data, up to 30% of all people over 85 suffer from Alzheimer’s disease, and 5% of people over 65 suffer from Parkinson’s disease [3]. This increase in the number of patients as well as the need to utilize a comprehensive approach necessitates the high standards for medical data systems and machine learning algorithms deployed on these systems [4].

### 1.1. The Assessment of Neurodegenerative Disorders

The effectiveness of a given therapy in slowing down the progression of neurodegenerative disorders depends on their timely detection. Early detection through screening will allow initiation of treatment before the onset of severe clinical symptoms and significantly delay their onset. In recent decades, pre-clinical assessment has become a major focus of research into neurodegenerative disorders [5,6]. The assessment might take into account behavioral symptoms, biomarkers from blood and cerebrospinal fluid, neuroimaging data (PET and MRI), conductivity and electrical activity of the brain (TMS and EEG), and the results of neuropsychological tests. Often, differential diagnosis is required for depression and conditions that could cause a potentially reversible cognitive decline [7]. Thus, the number of instrumentally determined markers and parameters is quite large. A feature of most neurodegenerative disorders is the low specificity of individual assessment methods and markers. This often leads to the inability to establish the correct diagnosis without multifaceted and often costly examinations [8]. Below, we will describe the main groups of diagnostic methods used both to detect the disease and to monitor its progression.

### 1.2. Biomarkers in Blood and Cerebrospinal Fluid

The main biomarkers used in the differential diagnostics of neurodegenerative disorders, in particular Alzheimer’s disease, are the levels of beta-amyloids and tau proteins in the cerebrospinal fluid of patients [9]. Using the ratio of the amount of beta-amyloid 1-42 to the total tau index makes it possible to identify patients with neurodegenerative disorders with high accuracy as well as to distinguish patients with Alzheimer’s disease from patients with other forms of dementia with moderate accuracy [10]. The level of tau protein phosphorylated at position P217 (P217 tau) in blood plasma also demonstrated the potential to be used in differential diagnostics of neurodegenerative disorders, primarily Alzheimer’s disease [11]. Tau 217 values outperformed other blood biomarkers, as well as MRI markers, in a study of 1402 patients from 3 independent cohorts that included patients with dementia, a cognitive decline of a different nature, and healthy subjects [12]. The accuracy of the assessment using blood biomarkers in this study was comparable to the accuracy of diagnosis using PET markers. Other studies have also demonstrated that tRNAs could serve as potential biomarkers to detect various neurodegenerative disorders at an early stage [13].

### 1.3. Psychological and Neuropsychological Assessment

Psychological (including experimental pathopsychological assessment as practiced in post-USSR countries) and neuropsychological methods for assessing the cognitive deficit present in neurodegenerative disorders are also used to monitor the progression of the disorder and the effectiveness of treatment. The most sensitive to specific cognitive deficits caused by neurodegenerative disorders are short-term memory assessments (e.g., the Visual Short-Term Memory Binding Test) and executive functions assessments. A meta-analysis of 142 trials of sorting tests evaluating executive functions (Wisconsin Card Sorting Test [WCST] and Delis-Kaplan Executive Function System [DKEFS-ST]) demonstrated their high diagnostic validity in detecting neurodegenerative disorders and vascular dementias [14]. The results of memory and executive function tests, conducted in isolation or as part of standardized neuropsychological batteries, correlate with pathological changes in the medulla as well as with the patient’s overall level of maladjustment. Their use, therefore, represents a cost-effective and non-invasive way to track the progression of the disease. Neuropsychological testing has also demonstrated the potential for the pre-clinical diagnosis of neurodegenerative disorders, especially Alzheimer’s disease. Composite score from the ADCS-PACC scale, which is derived from several neuropsychological tests of memory and executive functions, showed a correlation with the level of beta-amyloids in the cerebrospinal fluid of patients in two studies conducted in the USA and Australia [15]. These preliminary research results demonstrate that sufficiently sensitive neuropsychological batteries might be used as a non-invasive method for the early detection of neurodegenerative disorders.

### 1.4. Neuroimaging

The regularity of the spread of the degenerative process in the brain is one of the main criteria for the differential diagnosis of various neurodegenerative disorders. The formation of tau links, as well as a decrease in the volume of certain areas of the brain, correlates with the prognosis and clinical symptoms. These changes are determined using MRI and PET methods. MRI can detect damage to the white matter and a decrease in the local volume of various parts of the brain [16]. PET allows for assessing changes in glucose metabolism and the presence of neurofibrillary tangles in the brain [17]. Machine learning methods are currently actively applied to neuroimaging data to minimize diagnostic errors and automate the process. Currently, MRI and PET methods are quite expensive and their use as methods for pre-clinical diagnosis of neurodegenerative diseases is severely limited.

### 1.5. Connectivity

Transcranial magnetic stimulation (TMS) demonstrated some potential for differential diagnosis in patients with neurodegenerative disorders of various etiologies. In a blinded study of 273 patients (with 421 controls) with Alzheimer’s disease, Lewy body dementia, and frontotemporal dementia, the assessment of cortical connectivity with TMS was conducted [18]. The researchers then trained a prediction model in the form of an ensemble of the binary decision three classifiers. The model showed extremely high classification accuracy when discriminating between disorders. This study demonstrates the potential of combining TMS and machine learning methods in the non-invasive differential diagnosis of neurodegenerative disorders.

### 1.6. Electrical Activity of the Brain

Abnormal patterns of electrophysiological activity of the brain in neurodegenerative disorders result from several factors including disturbed functional connections between cortical zones, axon pathology, cholinergic deficiency, and others [19]. Typical EEG patterns used in screening have been described in the scientific and practical literature [20]. However, at the moment, the described EEG patterns have not demonstrated high diagnostic accuracy. Potentially, their diagnostic accuracy can be significantly improved through the application of deep learning methods to the analysis of neurophysiological data. Such studies are already being actively carried out [21].

Thus, the development of algorithms for pre-clinical non-invasive assessment of neurodegenerative disorders and their differential diagnosis remains a promising area of research. The main challenge is to ensure the generalizability and universality of developed algorithms, so they can by deployed using various EEG devices. The other promising direction of research is the application of machine learning methods to the differential diagnosis between neurodegenerative disorders and mild cognitive impairments of different etiology [22].

### 1.7. Application of Machine Learning

The process of establishing a diagnosis varies depending on the data about a patient available to a medical practitioner at a particular moment. The results of psychosocial and behavioral assessments are often presented as a small set of tabular data and/or several composite indicators. In this case, the practitioner makes a decision based often on his subjective clinical experience. The evaluation of clinical tests (biomarkers of blood, cerebrospinal fluid, etc.) is often also carried out by the medical practitioner who is provided with the generally accepted reference values. That is, for each set of parameters, there are specific ranges that are the “norm” for a healthy person and numerical indicators of the level of deviations associated with potential health problems. This form of data presentation is easy to interpret and convenient for the practitioner.

However, when interpreting the results of neuroimaging and the analysis of the electrical activity of the brain, not only the amount of data is increased substantially, but the format of the data presentation changes. Instead of concise tabular values, a medical practitioner sees images of nerve tissues or a graphical representation of the electrical signals of the brain. In this form, the data are hard to interpret by a person directly, and to establish a diagnosis, preliminary decoding based on complex groups of instrumental parameters is necessary. The situation complicates further if the assessments are performed periodically and there is a need to track the progression of the disease. In this case, the complexity of a patient’s data increases significantly along with the probability of a medical error, simply because of the large number of parameters examined. Therefore, although the use of neuroimaging (PET and SPECT in particular) is one of the most effective and accurate methods for detecting neurodegenerative disorders at the pre-clinical stage, these methods are rarely employed as part of routine medical screenings [23]. The problem of analysis of patient data that are too large in volume and too complex for human perception has a potential solution: automation of processing using mathematical pre-processing and machine learning technologies. Modern algorithms can already analyze large volumes of data and produce summary reports with a list of detected signs of a patient’s health problems. Of course, this method has disadvantages: difficulties in creating a training sample of medical data of sufficient size and quality, “opacity” of algorithm predictions (a medical practitioner cannot know why the machine made this or that decision), technological challenges (deployment of such systems in the infrastructure of a medical institution could be a difficult task), social perception (distrust of automatic “assistants” by medical practitioners and patients), and the presence of controversial ethical aspects regarding the use of such technologies in healthcare [24].

To date, diagnostic systems based on machine learning are rarely used in clinical practice, since the listed disadvantages outweigh their potential advantages. However, a large body of research and development in this area demonstrates that such advanced systems will play an important role in the delivery of palliative care for patients with neurodegenerative disorders. This statement is supported by a significant increase in the number of relevant publications: almost an order of magnitude over the period 2014–2019. It should be noted that at the same time, the proportion of research devoted to deep learning algorithms has noticeably increased, which is illustrated by Figure 1 [4].

In support of this argument, we will further report on some of the applications of machine learning to the assessment and correction of the disorders under consideration. Assistive technologies and methodological approaches will be discussed separately in the relevant sections below.

The use of machine learning for the analysis of blood and cerebrospinal fluid biomarkers is relatively uncommon, and the algorithms used, in most cases, are quite simple from a mathematical point of view. For example, the method of linear discriminant analysis (LDA) has been used to stratify patients by their blood work [25]. In total, 377 people participated in the study: individuals without symptoms of neurodegenerative diseases (97 individuals), with signs of Alzheimer’s disease (35 + 41 with mild cognitive impairment), with parkinsonism (57 + 29 with cognitive impairment in addition to the usual damage to motor functions), with dementia due to Parkinson’s disease (87), and with frontotemporal dementia (31). Five blood markers were assessed: the levels of beta-amyloids 42 and 40, the total level of tau protein (t-Tau) phosphorylated at position 181 (p-Tau181), and alpha-synucleins. LDA was used to build 2D and 3D linear models from blood work data, and subsequent classification was performed using a random forest (RF) algorithm. The average accuracy of differentiation by categories (Alzheimer’s disease, parkinsonism, and frontotemporal dementia) was 76%, and by subgroups: 83% in the spectrum of Alzheimer’s disease and 63% in the spectrum of Parkinson’s disease. The researchers in another study used cerebrospinal fluid to differentiate patients with Alzheimer’s disease from healthy controls [26]. They examined the levels of beta-amyloid 42, t-Tau, and p-Tau. The inflammatory proteins were determined by immunoassay. The authors used the LASSO algorithm (linear model with L1 regularization) for feature selection and logistic regression for classification, which reached 0.91 AUC (quantitative interpretation of receiver operating characteristic curve [ROC]). Another study used an alternative set of indicators from cerebrospinal fluid: the presence of elements in the fluid (As, Fe, Mg, Ni, Se, Sr, etc.). Their presence was determined with optical emission spectrometry systems with inductively coupled plasma and sector mass spectrometry. The data were then processed by a support vector machine (SVM) algorithm with a radial kernel [27]. As a result, the researchers were able to achieve an AUC of 0.76 for the diagnosis of Parkinson’s disease (binary classification versus healthy people) in a sample of 187 people (82 patients with parkinsonism). However, the out-of-sample prediction performance of the resulting model turned out to be quite unusual. At first, the model showed high accuracy when fitted on local patients and extremely low accuracy in people living in geographically remote areas. This was due to the characteristics of the training sample: SVM was trained on the sample of individuals living in the same area. After the model was re-trained using cross-validation with different territories serving as references, the average accuracy of the algorithm reached AUC 0.78, regardless of where the patients came from. The results of the latter work highlight the importance of applying proper training procedures and the need to use a sufficient data sample, even for relatively straightforward algorithms.

The diversity of changes in brain structures in neurodegenerative disorders, including those linked to genetic factors, makes creating computational models a challenge. The performance of the model can sharply decrease if the preliminary classification of pathological changes used to train additional models was performed erroneously. The possibility of automating pre-classification has been explored in detail by A. Young et al. [28]. The researchers made an attempt to combine algorithms that identify subtypes and the progression of neurodegenerative disorders into a single ensemble of learning models. In addition to a very detailed pre-classification of images by the type of brain changes, the authors created a model that estimated the stage of disease progression. To do this, they introduced time granulation related to the severity of a particular observed parameter within each pre-classified type of brain change. The model was also supplemented with data on genetically caused symptoms (Genetic Frontotemporal Dementia Initiative [GENFI]) that significantly affect the course of the disease. The mathematical model itself was an ensemble of simple linear z-score-based models that performed clustering tasks. For genetically determined frontotemporal dementia, the SuStaIn model was able not only to identify the genotype based on MRI images alone but also to resolve further heterogeneity within specific genotypes. For Alzheimer’s disease, the model was able to identify three subtypes with unique progressions over time. A more detailed specification of the structure of each subtype of disease according to physiological characteristics (e.g., localization and severity of atrophic changes) could be considered a fairly promising approach and, in addition to the work noted above, is covered in [29].

Peran and Barbagallo [30] compared MRI image classification algorithms for distinguishing cases of multiple system atrophy and Parkinson’s syndrome in a mixed dataset (29 patients with MSA and 26 with Parkinson’s syndrome) using multimodal MRI. Comparing the algorithms of discriminant analysis, voxel-based morphometry (supervised learning), and Kohonen networks (unsupervised learning), the authors concluded that the higher number of MRI modalities leads to better classification performance regardless of the processing method used.

For Alzheimer’s disease, the published studies [31,32] demonstrated the advantage of analyzing morphological and physiological changes in the brain with deep learning methods using MRI data. For example, Xinyang Feng et al. [31] compared the performance of a convolutional neural network to the classification based on morphometry markers, cognitive tests, and PET scans obtained from a multimodal dataset containing 975 images of patients with Alzheimer’s syndrome and 1943 images of healthy controls. The accuracy of the model based on the convolutional neural network turned out to be very high (AUC = 0.973). In addition, the developers were able to accurately determine specific anatomical features that turned out to be significant for the classification. It was due to the visually illustrative way of how information is represented in convolutional networks. This feature of convolutional networks was implemented in [32] as an independent diagnostic tool directly used by a medical professional. The researchers proposed not only to classify MRI data, but also to visually display the detected significant “abnormal” anatomical zones. Their relationship maps with morphological signs of atrophy correlated well with the results from earlier studies. This approach makes the application of the algorithm much more transparent for medical practitioners and increases confidence in its estimates.

To improve the speed of machine learning processing, Leonie Henschela et al. [33] proposed an algorithm for rapid segmentation of the entire brain into 95 classes using a convolutional neural network. The FastSurfer algorithm proposed by the authors has been commonly used as a means of preliminary data markup. This machine learning pipeline can be used as a standalone classifier and can also be integrated into more complex architectures.

There is an apparent lack of studies into the application of deep learning methods (when the model classifies “raw” data without access to human-designed features) to the analysis of blood and cerebrospinal fluid biomarkers. This can be explained both by the comparative simplicity of this data modality (a limited set of numerical values that is easily interpreted by a person), which eliminates the need to use “complex” solutions, and by insufficient research on this type of biomarker in the context of neurodegenerative disorder diagnostics in general. For example, in [34], a simple method for analyzing the concentration of p-tau181 in a patient’s blood is proposed, which provides accuracy and specificity sufficient for the preliminary screening of patients with suspected Alzheimer’s disease. However, the specificity of such a single marker is insufficient for a reliable final diagnosis, which makes it necessary to obtain additional tests. At some point, the amount of diverse data necessitates the use of more powerful machine learning methods.

It is important to note that the results of comparative studies show the advantages of deep learning models compared to simpler models in the diagnostics of various symptoms. The accuracy and specificity of deep learning models are higher than those of classical models [35].

An extensive comparative study of the accuracy and specificity of different algorithms for the detection of Parkinson’s disease was carried out by Mei et al. [36]. The authors concluded that the classification accuracy of machine learning algorithms was, on average, about ~94% for SPECT, ~86% for PET, and ~87% for MRI (including fMRI). The most used when working with neuroimaging data and the most accurate classifiers were SVM algorithms and artificial neural networks. SVM was used more frequently (50–70% for SPECT or PET and ~60% for MRI) compared to neural networks (22–53% for SPECT or PET and ~23% for MRI). Figure 2 demonstrates different data types and ML models described in the literature. 

Machine learning methods were also successfully employed in the studies into genetic markers of neurodegenerative disorders. For example, when studying miRNAs and non-coding RNAs, Ángela García-Fonseca et al. [37] noted the complex relationships of identified sites with specific disorders and outlined the possibility of modelling these relationships. The most effective algorithms were shown to be SVM and various types of decision trees. An example combining manual feature selection and deep learning was provided by Yuan Sh et al. [38]. As in the previous study, the authors confirmed, using statistical methods of data analysis, the presence of a strong correlation between miRNA expression and ratios of various blood cells. The authors selected the most informative features for their machine learning model. The selected features were associated with the presence of Alzheimer’s disease, mild cognitive disorders, and were also predictive of the disease’s progression. The final model was based on a non-linear neural network and demonstrated the out-of-sample classification accuracy of 91.59%. The area under the curve (AUC) for the control group, the mild cognitive impairment group, and the Alzheimer’s group was 0.97, 0.95, and 0.98, respectively.

Similarly, Jianhu Zhang et al. [39] suggested working with blood biomarkers. As in the studies described above, feature selection was performed manually based on statistical data. The neural network-based classifier demonstrated an accuracy of about 98%, which is roughly in line with the results shown in previous work.

The general mechanisms linking neurodegenerative and other mental disorders are important to consider for theories that aim to describe the relation between genetic markers and the progression of pathological changes in the brain [40]. The discovery of common mechanisms for neurodegenerative and mental changes might lead to new research avenues in omics technologies for diagnostics.

The growing popularity of computational approaches in bioinformatics, the relative simplicity of analyzing biological fluids statistically, and the good interpretability of the obtained results will likely lead to a steady increase in the number of studies on the assessment of neurodegenerative disorders using machine learning applied to genetic data. In addition, the possibility of using genetic data with instrumental and sociodemographic variables is attracting more and more attention [41]. 

The nature of the data obtained during the neuropsychological examination does not encourage the use of deep learning, so we will focus on classical algorithms. In one study, the use of 9 standardized psychological tests (including the assessment of memory, speech, daily activities, and general cognition) in combination with a test of spatial attention (also to evaluate cognitive abilities) in a sample of 28 patients with Alzheimer’s disease and 50 healthy controls made it possible to achieve a classification accuracy of 91.08% for the SVM algorithm [42]. The authors also evaluated several other algorithms: RF, gradient boosting (GB), and its variation adaptive boosting (AB). However, the performance of these algorithms was noticeably lower: 81.75%, 85.92%, and 80.75%, respectively. A slightly more sophisticated approach was implemented in another study, where the authors classified the patients into three groups (healthy controls, possible Alzheimer’s disease, and confirmed Alzheimer’s disease) with a two-stage algorithm based on additional modality data [43]. Initially, the authors identified patients at risk based on biomarkers: levels of cholesterol, blood glucose, high-density lipoprotein, glycosylated hemoglobin, and blood pressure, which was performed using SVM. In the second step, the results of a standardized cognitive ability test (CAT) of the selected individuals were classified by a polynomial logistic regression (LR) algorithm. The accuracy of the SVM classifier was 0.86 AUC, and the addition of LR increased it to 0.89 AUC; these results were achieved on a sample of 2361 patients.

Unfortunately, these and other similar studies are not of particular interest with regard to the objective diagnosis of neurodegenerative disorders. The data on the cognitive and psychological state of a person underlying their predictions have little specificity: similar symptoms (decreased cognitive abilities, lowered mood, etc.) are observed in various disorders with different etiologies. The more promising approach might be to combine data from neuropsychological assessments with data on pathophysiological changes in the brain (e.g., recorded using neuroimaging methods). This would allow not only to maintain the high sensitivity of neuropsychological data but also to significantly increase the specificity of diagnosis. For example, combining the results from the Addenbrooke’s Cognitive Examination, INECO Frontal Screening, and several parameters calculated from MRI data (in particular, volumes of gray matter in different areas of the brain) made it possible to carry out differential diagnosis of Alzheimer’s disease and frontotemporal dementia with an average accuracy of more than 0.91 AUC [44]. Moreover, despite the use of classic machine learning algorithms (the k-means method for primary feature clustering with LR for the final classification), this approach also demonstrated an outstanding generalizability: the model trained on patient data from one country (Argentina) retained an average accuracy above 0.9 AUC when validated on patients from other countries (Colombia and Australia). In another study, the authors examined different combinations of cognitive and neurophysiological markers to estimate which would perform the best for the differential diagnosis of Alzheimer’s disease (171 patients) and frontotemporal dementia (72 patients). They used a genetic machine learning algorithm for selecting optimal features for subsequent classification using SVM [45]. The final values of the f-measure (similar to AUC, but more reliable for data with a strong asymmetry in the distribution of class labels) were as follows: only for cognitive factors (memory, speech, attention, etc.), 0.882; only neurophysiological (hypermetabolism foci in various areas of the brain), 0.921; and the optimal combination of traits, 0.949.

The implementation of machine learning methods will likely reduce costs by enabling faster diagnostics as well as reducing the likelihood of medical errors. There are three main areas of application of modern methods of data analysis in the diagnosis of neurodegenerative disorders:

1.Increasing the accuracy and coverage of pre-clinical diagnostics. Identification of the pathological process before the onset of clinically observable symptoms will allow treatment to be started in advance, slowing down the progression of the disease and improving the quality of life of the patient.2.Differential diagnosis of various neurodegenerative disorders and comorbidities. Increasing the accuracy of diagnosis will improve the selection of treatment and correction of symptoms.3.Increasing the accuracy of forecasts for the progression of the disease after its onset. The application of machine learning methods to the analysis of longitudinal data will make it possible to build dynamic models for symptom monitoring.

### 1.8. The Treatment of Neurodegenerative Disorders

Due to the irreversible nature of the degeneration of the nervous tissue, the therapy of neurodegenerative disorders is aimed at slowing down the progression of degeneration and improving patients’ quality of life. Pharmacotherapy slows down degenerative processes through changes in the metabolism of neurons and glial cells. In addition, pharmacological interventions can be aimed at compensating for the functions of destroyed neuron populations. Non-pharmacological interventions are aimed at improving adaptation to the environment and improving the quality of life of the patient as well as helping patients to perform their household tasks. The main approach to non-pharmacological interventions is comprehensive rehabilitation that combines aerobic physical activity, cognitive training, and occupational therapy [46]. It is also possible to additionally use methods of brain stimulation.

### 1.9. Pharmacological Therapy

The main drugs used in the pharmacotherapy of neurodegenerative disorders are cholinesterase inhibitors (donepezil, rivastigmine), NMDA receptor antagonists (memantine), and their combinations [47]. Dopamine receptor agonists (apomorphine), dopamine precursors (levodopa), and monamine oxidase inhibitors (MAO-B) are used to reduce the intensity of motor disorders in parkinsonism (clinical syndrome) [48]. At the moment of publication, studies into potential neuroprotective drugs are being conducted, but there is no consensus on the use of such drugs in practice.

### 1.10. Cognitive Training

Exercises aimed at training memory, attention, and thinking have been extensively used as a therapy for individuals with neurodegenerative disorders [46]. Examples of such training include episodic memory capacity training with mnemonic techniques and visual information processing speed training [49]. The meta-analysis of 32 studies on the effectiveness of cognitive training in individuals with neurodegenerative disorders showed moderate effectiveness of training for improving cognitive functions [50]. However, many of the studies included in the meta-analysis were of low quality and had small sample sizes, so the authors cautioned against drawing definitive conclusions. At the moment, the effectiveness of cognitive training for the therapy of neurodegenerative disorders has not been sufficiently studied.

### 1.11. Physical Exercises

Decreased levels of neurotrophic factors—mainly brain-derived neurotrophic factor (BDNF) and its receptors—are one of the most common physiological consequences of various neurodegenerative disorders. The meta-analysis of 18 randomized trials showed an association of exercise with neurotrophic factor levels in patients with neurodegenerative disorders. Physical exercise increases the level of a neurotrophic factor in blood plasma [51]. High concentrations of neurotrophic factors reduce the toxic effect of nerve cell death [52,53]. In the studies, an increase in the levels of neurotrophic factors in plasma occurred regardless of the type of exercise: aerobic, strength, or combined exercise programs had similar effects.

### 1.12. Ergotherapy

Lifestyle counselling, socialization assistance, and re-training of everyday skills are critical for slowing down the progression of neurodegenerative disorders and improving patients’ general quality of life [54]. In some European countries and the US, occupational therapy is often provided by specialized employees of medical and social institutions (occupational therapists). In Russia, occupational therapy is usually provided by clinical psychologists.

### 1.13. Brain Stimulation

Methods of TMS and deep brain stimulation (DMS) have not demonstrated significant effects on clinical outcomes in neurodegenerative disorders [55]. One exception is the use of the DMS method for the correction of motor symptoms in Parkinson’s disease. The DMS stimulation led to a clinically significant decrease in tremor and muscle rigidity in patients.

### 1.14. Prevention of the Associated Psychological Problems

Affective symptoms (including anxiety and depression) can occur in patients with neurodegenerative disorders. The symptoms could occur as a direct result of dementia, and independently as a reaction to one’s disease or social situation (e.g., as a reaction to impaired communication with relatives) [56]. Psychosocial interventions could be aimed at facilitating social support and managing the symptoms of anxiety and depression.

Early pre-clinical diagnostics of neurodegenerative disorders will enable the early start of pharmacological therapy, which would slow down the degeneration of the brain matter, as well as cognitive training, which would slow down the development of cognitive deficits. Currently, pre-clinical diagnostics of neurodegenerative disorders using neuroimaging, brain stimulation, and analysis of brain electrical activity is the most promising area for the application of machine learning and data analysis methods.

### 1.15. Application of Machine Learning

Since the changes in brain matter that occur in neurodegenerative disorders are often irreversible, their therapy is aimed primarily at maintaining the quality of life of patients. Machine learning might be employed to assist with the therapy in several ways: the evaluation of the effectiveness of therapeutic interventions, dynamic monitoring of the disease progression, and the optimization of brain stimulation methods.

When conventional DBS therapy is conducted, continuous pulsation is delivered without adjustment for the patient’s current condition. This approach could lead to sub-optimal results. One possible way to optimize the delivery of pulsation would be to use real-time data on the patient’s condition to select the right moment for stimulation. These data can include information obtained from chemical sensors (e.g., dopamine levels), accelerometers, and ECoG and EMG data [57]. For example, the occurrence of tremors can be detected in real time by applying a pre-trained classifier to the EMG signal [58]. In the case of DBS, this approach is called adaptive DBS.

Other methods of pharmacological therapy and physiotherapy used in individuals with neurodegenerative disorders are less invasive compared to DBS. However, since these methods’ effects are not immediate, evaluating their effectiveness can be more challenging. This problem can be solved by adapting the approach used for diagnostics: automated gait analysis [59] using wearable devices or computer vision systems. A similar approach is used to prevent fall injuries by continuously monitoring and evaluating the probability of falling using a trained regression model [60]. The analogous solutions for the assessment of neurodegenerative disorders are discussed in the corresponding section. 

In conclusion, it is important to note the partial intersection of the proposed methods both in the feature space and in the algorithms being used. There is an emerging approach to aggregate individual markers and even intermediate diagnostic models into higher-level hierarchical ensembles. This approach entails an end-to-end system for processing instrumental data, whose results can then be used in different processes including diagnostics and disease progression monitoring (as depicted in Figure 3). 

Such an approach, however, is characterized by high computational, technological, and regulatory complexity. To develop such multichannel systems, it is also necessary to address the issues of collecting and pre-processing data, cross-checking the fit of models, and verifying the effectiveness of the analyses.

## 2. Depressive Disorders

Around 15% of the world population has been diagnosed with a depressive disorder at least once in their life [61]. The average annual prevalence of depressive disorders among the adult population, according to the World Health Organization, is 5% [62]. Depression is one of the most common mental disorders. Depression negatively impacts the ability to work, decreases the quality of life, and constitutes a major risk factor for suicide [63] and other adverse health outcomes [64]. Subclinical depression may also precede the onset of neurodegenerative disorders [65].

### 2.1. Diagnostics of Depressive Disorders

Depressive symptoms can have many etiologies, and their potential neurophysiological mechanisms have not been well understood yet. The current consensus is to consider depressive states as a consequence of poorly differentiated functional, neurotransmitter, and metabolic changes in the brain [64]. Since there is no specific localized pathological process in the nervous tissue in depressive disorders, neuroimaging methods for the diagnosis of depression have been rarely used in practice. Biomarker studies of blood and cerebrospinal fluid have been also rarely used. Diagnosis of major depressive disorder, minor depressive disorder, or bipolar depression is usually based on the results of clinical interviews and standardized questionnaires. The potential use of brain electrical activity data to diagnose depression and predict therapeutic responses to antidepressants is currently the subject of active research.

#### 2.1.1. Use of Standardized Questionnaires

In addition to the clinical interview, standardized questionnaires are used to assess the severity of depressive symptoms. In Russia, among others, the following are used for this: the Beck Depression Inventory (BDI) [66], the Hamilton Scale (HAM-D) [67], and the Hospital Anxiety and Depression Scale (HADS) [68]. The diagnosis of a depressive disorder is not made solely based on questionnaires. Their results serve as a guide for a psychiatrist in determining the severity of symptoms and finalizing the diagnosis. The final diagnosis is based on the combination of identified symptoms that fit the diagnostic criteria of the International Classification of Diseases. However, the periodic completion of questionnaires by the patient can be used as a relatively objective way to track the dynamics of depressive symptoms to adjust the course of treatment. 

#### 2.1.2. Analysis of the Functional Activity of the Brain (EEG)

Methods for analyzing brain activity can be used both to diagnose depressive states (in particular, to assess the severity of depression) and to predict the response to pharmacological treatment. For example, a decrease in interhemispheric coherence of the frontal leads, observed regardless of the etiology of the depressive state, has been identified as a potential predictor of resistance to pharmacotherapy [69]. Identification of predictors of drug resistance and response to drugs is one of the directions of research on the use of EEG methods in the diagnosis of depression [70].

#### 2.1.3. Application of Machine Learning

Attempts have been made to use neuroimaging data (MRI, functional MRI, and diffusion MRI) [71] and brain electrical activity (EEG) data [72] to diagnose depressive disorders. The recorded data were analyzed with classical machine learning algorithms (e.g., the support vector machine) and deep learning methods [73].

The main types of data sources and methods used in the assessment of various mental disorders, including depression, were described in the review by Chung and Teo [74]. The authors analyzed several works on the assessment of anxiety and depression based on written texts, voice recordings, data from MRI scans, survey methods, and their combinations. The paper noted that different algorithms were better suited for different types of data (they examined gradient boosting, random forest, and neural networks), but the average accuracy of the best-performing algorithms did not exceed 80%. The authors attribute the low results both to the small amount of data and to incomplete or imbalanced training datasets. Chen et al. [75] also noted the need to use additional data channels, including voice, activity, sleep, questionnaire, and instrumental methods (e.g., MRI) to potentially improve the accuracy of diagnosis. Additional data channels can also represent different aspects of everyday activity to add behavioral patterns into the data. The authors emphasized the promise of multimodal datasets, since various mental disorders are often accompanied by external signs progressing with time (Table 1). 

The use of multimodal data potentially makes it possible not only to diagnose a wide range of mental disorders, but also to monitor the dynamics of the development of the disease, since the observed signs become more pronounced at later stages. However, the collection of multimodal data is often associated with a number of difficulties, in particular with the need to observe the patient for a long time, as well as to confirm the observed signs by cross-checking on historical data performed by an experienced doctor. Big data collected from mobile devices reflect the patient’s social interactions and fine motor skills features such as taps duration, typing speed and rhythm, reaction speed, etc. It is worth noting that in order to automatically collect such an amount of data, the mobile application installed on the patient’s device should have access to the correspondence and system software, which can become a security issue. However, such solutions can be easily implemented on the architecture of social interaction platforms such as social networks. The authors proposed the implementation of platform solutions for big data as the main processing unit. The units then are used as a basis for more complex models. In addition, the authors proposed a machine learning model that can be used to evaluate the effectiveness of treatment and to monitor the patient’s condition using biofeedback.

One of the features of mental disorders, when compared to other disorders, is that their assessment relies heavily on the subjective experience of a patient subjectively described by him or her. The quality of the assessment is highly dependent on the clinical experience of the medical professionals conducting the assessment. To systematize the data collection process of mental illness clinical signs, a general methodology for filling the database is needed. The same methodology, on the other hand, forces doctors to use standardized data collection forms, which positively affects the entire dataset. The fact is that the different experiences of doctors and the peculiarities of the method of presenting the picture of the disease cause certain markup anomalies in the database [76]. Often, this feature does not allow one to reasonably assert that the sample data are not biased. In addition, the amount of relevant data differs from channel to channel, which often requires the deployment of different cleaning and pre-processing algorithms.

In their review of machine learning methods for neuroimaging, Quaak et al. [77] pointed out the need for a more conservative approach to testing the quality of trained models, since their results on the test set often do not reflect the model’s real performance. In addition, the authors noted the popularity of the EEG as a source of data for the diagnosis of mental disorders. MRI/EEG databases for diagnostics of depression, which are used to train machine learning algorithms, can be recorded in different modes: when a patient is performing certain tasks (game), responding to external stimuli (video, images, and music), and in a resting state. The most common approach is to receive data from a person in a resting state (eyes open or closed).

The information about the patient’s depression is often considered protected private information, which leads to very few datasets being openly available. The information on openly available datasets for diagnostics of depression by EEG is presented in Table 2.

The common problem with openly available datasets is that they often include a limited number of participants (the average sample size of patients with depressive disorder is around 30 people). This problem calls into question the diagnostic value of studies conducted using the datasets. This problem can be mitigated either by combining existing datasets [82] or by the collection of more data. The latter option requires substantial resources.

Convolutional neural networks are the most common types of neural networks for classifying the presence or absence of a depressive disorder. They rely on a two-dimensional or one-dimensional convolution operation, or hybrid models of convolutional and recurrent neural networks [83].

The input of the neural network is either the raw EEG signal (after pre-processing in the form of filtering and denoising) or its converted version. Given the existence of well-known models of neural networks used for image classification, the transformation often consists of the formation of an “image” from the EEG signal, which is then fed to the input of a two-dimensional convolutional network. To form an “image”, the power values of brain rhythms can be used, which are spatially projected onto a plane, following the location of the electrodes [84].

Since deep learning algorithms are demanding towards an amount of training data, and the existing samples of training data are often of limited size, the augmentation method is applied. Augmentation is a method of artificially increasing the amount of data used for training [85]. The use of data augmentation in training deep neural networks can reduce the effect of overfitting and improve accuracy and stability. EEG data augmentation uses sliding window data sampling, data from generative models, noise addition (generally Gaussian noise is used), sampling, segment recombination, and Fourier transform [86]. The architectures of neural networks and the achieved accuracy of depression detection according to EEG data are presented in Table 3.

It should be noted that the claimed high accuracy of depression classification, exceeding 90%, may be associated with testing models on a limited set of initial data and possible incorrect partitioning of training data by patients, which might have led to the implicit leakage of training data into the test subset. Leakage of this kind leads to overestimated accuracy and reduces the generalizability of the model due to the effect of overfitting [99].

The application of methods for automated assessment of depressive disorder severity involves the selection of relevant features of the received signal and the development of complex metrics of the depressive state based on them. Isolation of non-linear spectral characteristics of EEG signals in combination together with the support vector machine methods demonstrated high classification accuracy in several studies [100]. The combination of linear discriminant analysis and genetic algorithms also demonstrated high discrimination performance when classifying patients with the depressive disorder [101]. This line of research can be developed further by incorporating new diagnostic categories as comparison groups and the development of classifiers with higher ecological validity.

The development and validation of composite diagnostic indices, as well as ensembles of algorithms for solving specific diagnostic problems, are most often carried out on small samples in the absence of independent external validation. This creates a high risk of overfitting the algorithms and reduces the generalizability of the results. To increase the validity in the development of EEG-based diagnostic indices, attention should be paid to the composition of a training sample, its size, and its quality. The use of the combination of neurophysiological data labelled according to a single protocol from different clinical sites could be an optimal solution.

### 2.2. Treatment of Depressive Disorders

The main methods of treatment of depressive disorders include pharmacological therapy and psychosocial interventions (psychoeducation and group and individual psychotherapy) [102]. Biofeedback and TMS are less commonly used. In individuals with depression, relevant therapeutic targets include maladaptive thoughts and beliefs, lowered mood, quality of life, and others. Reducing the risk of suicide in severe depression is also an important therapeutic goal.

#### 2.2.1. Pharmacological Therapy

The most common medication used for treating depression is serotonin reuptake inhibitors (e.g., sertraline and paroxetine), tricyclic antidepressants (citalopram and fluoxetine), and monoamine oxidase inhibitors (moclobemide and pirlindole). The mechanism of action of antidepressants is based on a change in the concentration of neurotransmitters available for binding in the brain, which leads to long-term potentiation or depression of synaptic connections [103]. The choice of medication is made based on their safety and the observed therapeutic effect in a particular patient. Doses are adjusted by the medical professional during treatment. Antipsychotics can also be used in depressive states that have arisen as part of psychotic disorders (e.g., bipolar disorder) [104]. Pharmacological therapy is usually only required for moderate to severe depressive disorders. The combination of pharmacological therapy and psychotherapy is optimal for the treatment of such forms of depression [105].

#### 2.2.2. Psychosocial Interventions

The main psychosocial intervention for working with patients with mild depression is psychoeducation, providing information about the symptoms and the possible progression of the disorder, recommendations for self-help, and lifestyle changes (if necessary). Other suitable psychosocial interventions for patients with depressive disorders are group and individual psychotherapy. The most studied approaches for the treatment of depression are cognitive behavioral therapy and its variations, psychodynamic therapy, schema therapy, and decision-oriented therapy [106,107]. In general, psychotherapeutic approaches do not differ significantly in effectiveness, so the choice of a particular approach in each case depends on the availability of a specialist and the individual preferences of the patient. Besides psychotherapy, regular physical exercise could also be considered a part of a psychosocial intervention. Regular exercise has been shown to be effective in reducing depressive symptoms [108]. Positive changes may result from the metabolic and hormonal changes that accompany regular exercise.

#### 2.2.3. Biofeedback

Biofeedback methods have also been shown to be effective in reducing depressive symptoms. A meta-analysis of 14 randomized controlled trials in 794 subjects showed significant improvements in symptoms in patients with depression [109]. Neurofeedback using EEG and fMRI demonstrated promising results in reducing depressive symptoms, but the number of published studies is still limited [110,111].

#### 2.2.4. Brain Stimulation

Studies demonstrated that transcranial magnetic stimulation had a potentiating effect on antidepressant intake and was associated with improved clinical outcomes [112]. TMS is recommended by several national medical agencies for use in the treatment of depression as a procedure with potentially positive results and no side effects [113]. However, there is currently insufficient data to draw definitive conclusions about the effectiveness of TMS in reducing depressive symptoms.

#### 2.2.5. Application of Machine Learning

Machine learning is also used to assess the treatment effectiveness in depressive disorders [114]. The effectiveness of depression treatment using antidepressants or TMS can be assessed by using and applying classical machine learning methods (support vector machine [115] and random forest [116]) or deep neural network [97] algorithms to EEG data [117].

The machine learning methods used and the accuracy achieved in predicting depression treatment based on EEG data are presented in Table 4.

Studies on the predictive treatment of depression, as well as studies on the diagnosis of depression, are characterized by a limited number of participants, which does not allow us to assert that the methods used have sufficient generalizability. At the moment, the most promising areas for the development of applied methods of machine learning for the therapy of depression are dynamic monitoring of the symptoms and treatment of pharmacoresistant depression.

## 3. The Use of Auxiliary Indicators

Along with well-established instrumental methods, such as MRI, fMRI, and EEG, it became possible to automatize the use of several auxiliary diagnostics indicators. These indicators had been used as visually observable markers of a disorder and were usually determined by a medical professional during direct contact with a patient. At the same time, the evaluation of the severity of these indicators and their association with the symptoms were always performed by medical professionals based on their subjective experience. Such auxiliary indicators include both observed disturbances in motor functions (gait, gross and fine motor skills, the ability to speak and swallow, etc.) as well as their various combinations with cognitive impairments. Automating the processing of such indicators is a promising direction of work for machine learning specialists since relevant training datasets are easy to collect and label, e.g., based on neuroimaging methods. For some types of motor disorders, researchers have already obtained promising results [120,121,122,123].

Instrumental methods for determining gait disorders, which are quite pronounced, in particular for Parkinson’s disease cases, are based on data from video cameras and sensors of wearable devices. Wearable devices are relatively easy to construct, and it is possible to use them outside the clinical setting to collect more data. On the other hand, small-sized three-axis accelerometers and inertial systems most often used for such solutions are subject to external interference because the device is loosely attached to the body and interacts with clothing and furniture. The daily activity data obtained from such devices are poorly interpreted outside the context of the situation. Nevertheless, the results of comparative experiments using such devices and machine learning methods look quite promising.

For example, Dante Trabassi et al. [122] provided a comparative analysis of various machine learning algorithms on a dataset from inertial sensors of a wearable device to detect gait anomalies which are common for Parkinson’s disease. The trials were conducted with 81 subjects with Parkinson’s disease and a control group of 80 subjects. The test protocol was the same for the test and control groups and included walking along a building corridor which was about 30 meters long with a wearable device in the form of a sensor on the belt. The collected data was pre-processed manually and then used to train several different machine learning models, such as decision trees, random forests, k-nearest neighbors, support vector machines, and artificial neural networks. The best result with an accuracy of 0.86 was achieved using an algorithm based on the support vector machine. At the same time, artificial neural networks showed a lower accuracy, comparable to a random forest.

In contrast, in an article by Robbin Romijnders et al. [123], the best result (classification accuracy of about 98% and classification completeness >95%) was shown using artificial neural networks. In this publication, a different approach to sensor placement was used, and gait features were reconstructed based on data from accelerometers and position sensors placed on the lower leg and ankle. The study used a larger dataset of 157 subjects with several disorders (Parkinson’s disease, multiple sclerosis, recent stroke, and chronic low back pain) and a larger test program that included the passage of a five-meter segment at a randomly defined slow, medium, and fast pace. Deep learning using temporal convolutional networks was chosen as the primary algorithm for this research. The high result shown allows us to conclude that this approach is suitable both for the preliminary screening of patients and for monitoring the dynamics of disease development.

Jinglin Sun et al. [120] proposed a similar approach for the diagnosis of Alzheimer’s disease, but eye movements were used as the primary source. A comparative study was conducted for classical machine learning methods (support vector machine and k-nearest neighbors) and well-established neural network architectures VGG-16 and Resnet-18, and for the architecture proposed by the authors based on a dual autoencoder module with a separate final classifier in the form of a fully connected three-layer network. Mean accuracy values of 0.87 ± 0.04 and recall values of 0.89 ± 0.04 were demonstrated, respectively.

Chonghua Xue [124] proposed to use and analyze patients’ voice recordings by convolutional and recurrent neural networks for the detection of Alzheimer’s disease. This architecture is well suited for time-distributed data; therefore, it is often used in voice analysis.

A number of works proposed an analysis of behavioral responses for the diagnosis of depression [125,126]. In such experiments, a patient is asked to determine the emotional coloring of various words and images that are demonstrated by a special program. The accuracy of the classifier based on bagged decision trees, trained on a sample with manual feature selection, reached 0.76, which does not allow us to speak of a stable result.

The issue of predicting and early diagnosis of cognitive dysfunctions in the elderly using machine learning methods on various data, including sociodemographic data, electronic medical records, clinical and psychometric studies, and neuroimaging data, is considered in a review paper [127]. Sarah Graham et al. highlighted the need to use multimodal datasets and maintain the transparency of the algorithms. They also noted the direct comparative analysis of the effectiveness of clinical methods for the early diagnosis of cognitive impairment and new methods based on machine learning have not yet been carried out.

A growing number of studies on the potential application of machine learning highlights the need of a general approach to evaluate such potential applications. Without determining the existing barriers to the introduction of technology, its features, strengths, and weaknesses, and understanding the methods for ensuring its safe use, the implementation of such methods in clinical practice will be difficult [128]. For most countries, the procedures of state regulators regarding the introduction of new medical instruments, as well as ethical issues related to the low transparency of the results of the model and the specific features of collecting and processing medical data, significantly slow down the introduction of systems based on machine learning. This was especially evident during the COVID-19 pandemic when many predictive, diagnostic, and clinical models were created, but only a few point developments reached successful practical use in medical practice. From a technical point of view, when developing solutions based on machine learning, it is necessary to carefully check all the main stages of data collection and preparation, justify the choice of a specific algorithm for solving the problem, and explain the results obtained [129].

Machine learning methods are gradually becoming an increasingly mature research tool in areas such as radiology, immunology, and the synthesis of new drugs [130]. There are studies on the application of machine learning methods in general, and in relation to the tasks of diagnosing neurodegenerative and mental diseases. In particular, there are a number of inter-related works that are being actively studied.

The issue of achieving consensus between several experts when labeling data and the features of training models on such a set is covered in [131]. The authors managed to get the convolutional neural network to reproduce the manual data markup features that are specific to each expert. Thus, the algorithm imitates the consensus of experts in the form of an ensemble of layers trained on separate markups.

The phenomenon of sustainable overestimation of the effectiveness of machine learning methods developed for use in psychiatry was studied in [132]. The authors noted that in the Predictive Analytics Competition (PAC-2018), in which participants were asked to identify depression from MRI scans, the results were between 60% and 65% on a large independent dataset, while on smaller datasets, the accuracy reached 80% or more. A more detailed study showed a steady overestimation of the accuracy of the model when testing on small sample sizes. For simple methods, in which the number of hyperparameters is small, with the correct tuning, the model is trained quite quickly, and with an increase in the size of the training sample, no further noticeable increase in accuracy and recall is observed. At the same time, many researchers split the dataset in such a way that as much of the data as possible is used in the training process. Given the relatively small number of publicly available datasets for training, the question of the realism of the declared high performance of the developed models when tested on a new larger independent dataset becomes increasingly relevant.

Based on the foregoing information, machine learning methods in no way can be considered a universal recipe for a diagnostician or therapist. A systematic approach has yet to be formulated and, at the moment, there are a number of shortcomings that are characteristic of not fully formed methodologies. Functionally, the first group can be attributed to “data and methods”, and the second to “the interpretability and trust”.

First, there is no generally accepted method for collecting information on the course of neurodegenerative diseases, on the basis of which it would be possible to collect a uniform and consistent dataset. As a result, available datasets described in the literature are often fragmentary and unbalanced or contain a limited number of respondents. The average sample size in terms of the number of respondents is about 500 people. The datasets themselves may contain more detailed information about each patient, but for a relatively small number of respondents. Thus, the dataset in [27] was collected from 277 patients. The largest dataset by the number of patients includes several thousand images but belonging to different patients [31]. It is worth noting the difficulty of tracking disease development because of data format limitations since they reflect the clinical situation only at a specific point in time.

To use the data in training models, it is required to clean them, balance by classes, and correctly split them into training and test sets. In addition, manual feature engineering based on the experience of the investigator and known patterns becomes important. Thus, in heterogeneous datasets, the features are difficult to validate and augment with data from other patients to increase the number of features. Augmentation with synthetic data is a promising and effective method, applicable for image [85] and time-series EEG data [86]. The selection of features for machine learning also depends on the characteristics of the dataset. A comparative study of some open datasets is given in [114]. Comparison of datasets, model’s accuracy, and participant-to-feature ratio is shown in Figure 4.

After data preparation, the model learns on the samples in the training set. As data differ in structure and volume in different sets, selection of a machine learning model becomes a non-trivial task. In particular, it is difficult to control the quality of learning in small datasets. In addition, the generalization ability of models is often overestimated due to the small sample size and possible overfitting, which was described in detail above. As a result, researchers offered a large number of models, with different architecture and characteristics. Such a cardinal difference does not allow for an adequate quantitative comparison for ensembled models, despite the simplicity of models such as decision trees or k-nearest neighbors.

Su et al. in [41] confirmed the complex nature of data mining and processing task for genetic and transcriptomic data on PD. The authors summarize the remaining limitations and challenges, and accordingly discuss potential future directions which may lead to promising machine learning approaches to address the discovered issues. Table 5 demonstrates several common points about dealing with mixed datasets and additional methods, applied in sequence both for the data and the model. The separate point on the interdisciplinary issue highlights the need for better collaboration and better domain expertise, as an expert’s opinion is still often the best ‘ground truth’ source. 

The issues of interpretability and trust to algorithms, listed in Table 5, can be broadened in several aspects. First, the regulatory framework for the use of machine learning models in clinical practice is rather incomplete. This is inherent not only in the diagnostics and treatment of neurodegenerative diseases, but rather in the clinical approach as a whole. As noted in [76], the methodology for processing mental disorders data for precision medicine approach is only being formed and it will take time to develop it. Second, whenever we use an artificial intelligence approach in healthcare, it is always related to making a decision. Disease detection, disease prediction, drug repurposing, precision medicine, medical resource allocation, and much more can be shown as evidence for it. In all of them, a machine learning algorithm either makes a decision or at least supports a decision. The system works as a complex visualization and analyzing tool to catch patterns in medical data [129]. The true reason why the recommendation system supports one decision or another is often hidden from the user. So, the questions of ethics and potential software issues arise and stay unresolved before medical systems become more mature and common in clinical practice. Quality and integrity checks of data, models, and protocols can help to speed up the development of ML-based medical systems and discover potential faults during development and trial stages. 

In a study by Vinny et al. [133], the authors critically examined the main issues of quality assessment and the possibility of implementing solutions based on machine learning models. The authors noted the need for a consistent multistage analysis of each of the components of the solutions offered to physicians, starting with the dataset, its size, completeness, representativeness, method and protocol of collection, and ending with the re-checking of the demonstrated characteristics of the model and the method of their determination. Separately, the authors noted the importance of transparency, the interpretability of the applied model, and the “common sense” check. The authors formulated the main provisions in 14 questions as shown in Table 6. 

## 4. Conclusions

The promising areas of research outlined in this review are currently being explored by various scientific groups around the world. The main factor limiting research in the identified areas is the lack of data of adequate quality and volume for training and validating machine learning models. The introduction of common standards for collecting, labeling, and storing data in shared databanks will allow the development of methods for the early diagnostics and correction of neurodegenerative and depressive disorders.

## Figures and Tables

**Figure 1 diagnostics-13-00573-f001:**
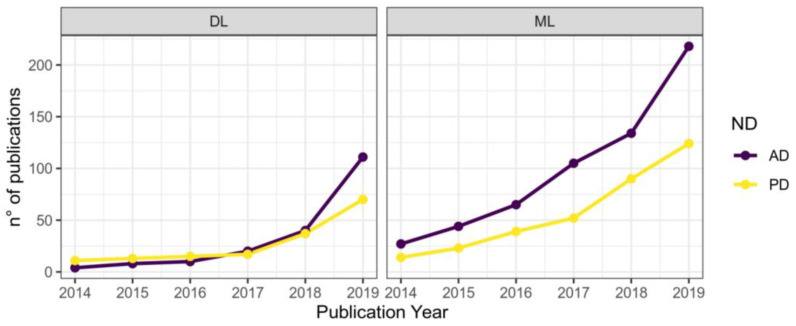
The number of publications on the applications of machine learning (**right**) and deep learning (**left**) to diagnostics of Parkinson’s disease (yellow graph) and Alzheimer’s disease (purple graph) by year (reproduced from [4]).

**Figure 2 diagnostics-13-00573-f002:**
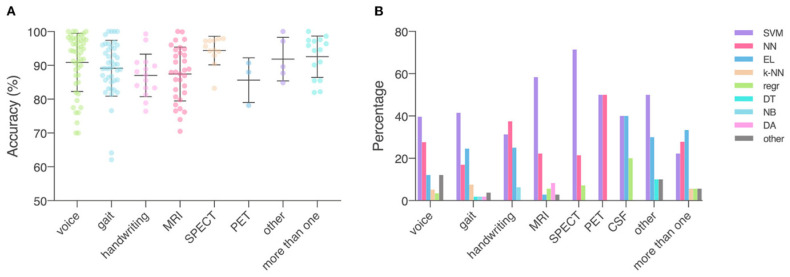
Data type, machine learning models applied, and accuracy. (**A**) Accuracy achieved in individual studies and average accuracy for each data type. Error bar: standard deviation. (**B**) Distribution of machine learning models applied per data type. MRI, magnetic resonance imaging; SPECT, single-photon emission computed tomography; PET, positron emission tomography; CSF, cerebrospinal fluid; SVM, support vector machine; NN, neural network; EL, ensemble learning; k-NN, nearest neighbor; regr, regression; DT, decision tree; NB, naïve Bayes; DA, discriminant analysis; other: data/models that do not belong to any of the given categories (reproduced from [36]).

**Figure 3 diagnostics-13-00573-f003:**
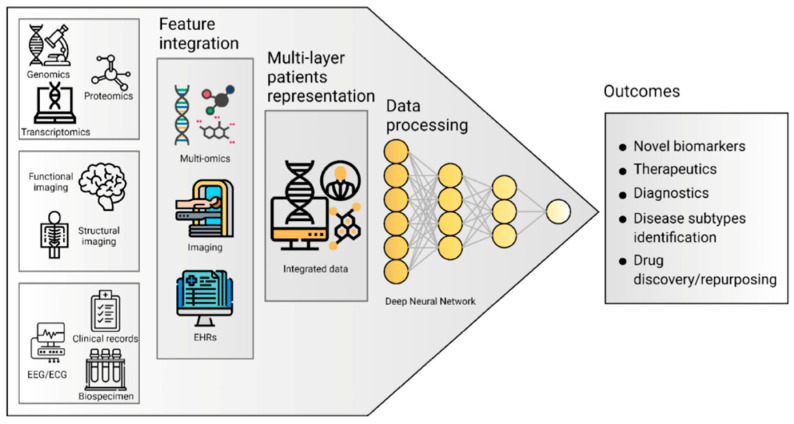
A multilevel approach to working with data on neurodegenerative disorders [4].

**Figure 4 diagnostics-13-00573-f004:**
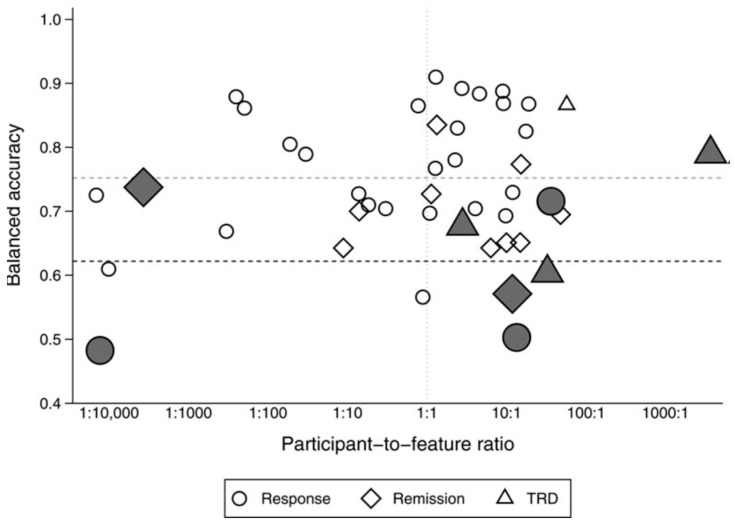
Balanced accuracy and participant-to-feature ratio in published machine learning studies of outcome prediction in the treatment of MDD. The x-axis plots the ratio of participants to predictive features. The y-axis plots the mean balanced accuracy within each study. Studies predicting response, remission, and TRD are plotted as circles, diamonds, and triangles, respectively. Adequate quality studies are highlighted with large, filled symbols. The dark gray horizontal dashed line shows the mean balanced accuracy of the eight adequate quality studies. The pale gray horizontal dashed line shows the average balanced accuracy of the other 45 studies (reproduced from [114]).

**Table 1 diagnostics-13-00573-t001:** Multimodal data features and their uses in mental health [75].

Features	Example(s)	Example Relevant in Mental Disorder(s)
Acoustic		
Source of sound features	Jitter	Increase with depression severity
Filtering features by vocal and nasal tracks	First resonant peak in the spectrum	Increase with bipolar severity
Increase with bipolar severity	Mel frequency cepstral coefficients	A variety of disorders
Prosodic features of speech	Pause duration	Higher in SCZ
Video		
Facial	Smile duration, eyebrow movement, disgust expression	Increased disgust expression in SI
Eyes	Gaze angle	More non-mutual gazes in MDD
Gait	Arm swing and stride	Reduced arm swing in MDD
Posture	Head pitch variance, upper body movements	Reduced head movement in SCZ Higher head movement in ASD
Language		
Grandiosity	Unrealistic sense of superiority	Increased in bipolar
Semantic coherence	Flow of meaning	Decreased in psychosis
Rumination	Repetitive thought patterns	Increased in MDD
Self-focus	Self-referent information	Increased in stress

**Table 2 diagnostics-13-00573-t002:** Available open datasets for diagnostics of depression by EEG.

N	Country	No. Participants	No. with Depression Disorder	Source
1	Malaysia	64	34	[78]
2	USA	121	46	[79]
3	China	53	24	[80]
4	The Netherlands	1274	426	[81]

**Table 3 diagnostics-13-00573-t003:** EEG classification accuracy for depressive disorder.

N	Architecture	Accuracy	No. Participants (Depression + Controls)	Source
1	1D CNN (9 layers)	98%	33 + 30	[87]
2	Hybrid 1D CNN + LSTM	96%	33 + 30	[87]
3	1D CNN (15 layers)	94%	15 + 15	[88]
4	Hybrid 1D CNN + LSTM	98%	15 + 15	[89]
5	1D CNN (5 layers)	96%	30 + 30	[90]
6	Hybrid 2D CNN + GRU	89%	24 + 29 16 + 16	[84]
7	2D CNN (8 layers)	99%	34 + 30	[91]
8	1D CNN (18 layers)	99%	15 + 18	[92]
9	Hybrid CNN + LSTM (6 layers)	99%	21 + 24	[93]
10	2D CNN	86%	24 + 27	[94]
11	Hybrid 1D CNN + LSTM	99%	34 + 30	[95]
12	Hybrid 1D CNN + LSTM (12 layers)	99%	46 + 75	[96]
13	2D CNN (8 layers)	99%	34 + 30	[97]
14	2D CNN (ResNet-50)	90%	46 + 46	[98]
15	2D CNN (8 layers)	68%	122 + 123	[99]

**Table 4 diagnostics-13-00573-t004:** EEG predictive accuracy of treatment for depressive disorder.

N	Type of Treatment	Accuracy	No. Participants	Source
1	Antidepressants	99%	17	[97]
2	Antidepressants	96%	30	[118]
3	Antidepressants	79%	122	[115]
4	Antidepressants	78%	51	[116]
5	TMS	82%	50 + 24	[117]
6	TMS	91%	46	[119]

**Table 5 diagnostics-13-00573-t005:** Summary points of challenges and potential future directions to address them [41].

Challenges	Potential Future Directions
Bias of sample size	Integrated multiple cohort modeling.
Handling whole spectrum genetic information	Engaging appropriate feature engineering tools such as genetic principal component analysis, multidimensional scaling, linear discriminant analysis, etc.; Incorporating appropriate deep learning model such as autoencoder.
Multifactorial modeling	Multivariate modeling; Incorporating kernel approaches and probability models.
Cohort diversity	Validation on an external cohort; Training model on data from multiple populations if possible; Engaging transfer learning.
Model interpretation	Using interpretable models such as Bayesian, rule-based (e.g., decision tree and random forest), logistic regression models, etc.; Incorporating or developing model interpretation methods for “black box” models, e.g., deep learning models.
Model evaluation	Evaluation using isolated validation dataset; Applying experimental test evaluation; Developing visualization tools for model evaluation.
Interdisciplinary issue	Deep interdisciplinary collaboration; Incorporating domain knowledge in model training.

**Table 6 diagnostics-13-00573-t006:** Summary of key questions in critical appraisal of a machine learning research paper [133].

№	Question
1	Was the study prospective or retrospective, observational, or randomized controlled trial?
2	Was the protocol published a priori?
3	Why was the dataset obtained, and what is its size?
4	What is the intended use of ML model in the context of the clinical pathway?
5	Does the dataset represent the disease spectrum in the target population?
6	How was the data split between training, validation and external testing?
7	How was the gold standard determined?
8	What was the type of ML employed? Which version of the model was used for the study?
9	Is the reported model “continuously evolving” or “continuously learning” by design?
10	Does the model suffer from a black box problem?
11	Which performance metric is being reported/optimized? How were performance errors identified and analyzed?
12	Is the model performance too good to be true?
13	Is the study repeatable and reproducible? Are source code and datasets available for scrutiny?
13	Does the AI intervention affect patient outcomes?

## Data Availability

No new data were created or analyzed in this study. Data sharing is not applicable to this article.

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
