# Peer review of "Modern Methods of Diagnostics and Treatment of Neurodegenerative Diseases and Depression"

_diagnostics, 2023, doi:10.3390/diagnostics13030573_

Round 1
Reviewer 1 Report
1. Lines 24-25 appear to lack the necessary punctuation, please add in.
2. There are serious errors in the References section, specifically the presence of duplicate serial numbers.
3. non-English characters (Russian) appear in the horizontal and vertical coordinates of Figure 1, please change them to English.
4. The vertical coordinates in Figure 1 lack numerical units, please indicate the numerical units.
5. The article should list relevant machine learning studies with necessary analysis and indicate solved or unsolved problems.
6. The article has too few graphs and charts. It is recommended that graphs and charts be drawn when summarizing relevant studies to enhance readability.
7. The authors may consider adding a section comparing machine learning techniques used in the diagnosis and treatment of neurodegenerative diseases and depression from a methodological perspective to highlight the technical summary of this review.
8. This review should address the limitations of existing machine learning approaches implemented in the prevention and correction of neurodegenerative and depressive disorders, pointing out where machine learning or deep learning methods need to be improved and the pressing problems that need to be addressed in the field.
Author Response
Thank you for reviewing the submission and your thoughtful comments on this paper. We value
your suggestions for improving the manuscript and appreciate this opportunity to address your comments and suggestions.
All the following points were addressed in the new version of the manuscript.
Point 1: “Lines 24-25 appear to lack the necessary punctuation, please add in.”
One missing punctuation mark was added.
Point 2: “There are serious errors in the References section, specifically the presence of duplicate serial numbers.”
References section was carefully revised and the reference numbering has been corrected.
Point 3: “non-English characters (Russian) appear in the horizontal and vertical coordinates of Figure 1, please change them to English.”
The axis label text in Figure 1 was changed to English.
Point 4: “The vertical coordinates in Figure 1 lack numerical units, please indicate the numerical units.”
As indicated by the new label text, the vertical axis in Figure 1 is the “number of applications” with no multiplier, thus no further corrections were introduced.
Point 5: “The article should list relevant machine learning studies with necessary analysis and indicate solved or unsolved problems.”
New text on the proposed topic was added on the lines 301–310, 557–577 and 585–589.
Point 6: “The article has too few graphs and charts. It is recommended that graphs and charts be drawn when summarizing relevant studies to enhance readability.”
Following the suggestion, new figures and tables were incorporated to improve readability. Particularly, Figure 2, Figure 4 and Table 1 were added, Table 5 was further expanded.
Point 7: “The authors may consider adding a section comparing machine learning techniques used in the diagnosis and treatment of neurodegenerative diseases and depression from a methodological perspective to highlight the technical summary of this review.”
Text on the proposed topic was added on the lines 837-910.
Point 8: “This review should address the limitations of existing machine learning approaches implemented in the prevention and correction of neurodegenerative and depressive disorders, pointing out where machine learning or deep learning methods need to be improved and the pressing problems that need to be addressed in the field.”
This topic was briefly addressed in the lines 837-910. Additionally, Table 5 was expanded in order to reflect relevant information.
Reviewer 2 Report
Interesting topic, good methodological approach.Very well written.
Author Response
Thank you for reviewing our manuscript. Your comments are very much appreciated.
Reviewer 3 Report
Reviewer comments
This is an interesting article on Modern methods of diagnostics and treatment of neurodegenerative diseases and depression with accepts for publication.
Comments
1- The English used correct and readable.
2- The work had a significant contribution to the field.
3- The work was well organized and comprehensively described.
4- There were appropriate and adequate references to related and previous work.

Author Response

(The authors gave the same response as above.)

Round 2
Reviewer 1 Report
I have no comment